# Synthesis of Silica Membranes by Chemical Vapor Deposition Using a Dimethyldimethoxysilane Precursor

**DOI:** 10.3390/membranes10030050

**Published:** 2020-03-22

**Authors:** S. Ted Oyama, Haruki Aono, Atsushi Takagaki, Takashi Sugawara, Ryuji Kikuchi

**Affiliations:** 1College of Chemical Engineering, Fuzhou University, Fuzhou 350116, China; 2Department of Chemical Systems Engineering, The University of Tokyo, 7-3-1 Hongo, Bunkyo-ku, Tokyo 113-8656, Japan; aaa.ooo.nnn.ooo@gmail.com (H.A.); atakagak@cstf.kyushu-u.ac.jp (A.T.); sugawara@chemsys.t.u-tokyo.ac.jp (T.S.); rkikuchi@chemsys.t.u-tokyo.ac.jp (R.K.); 3Department of Chemical Engineering, Virginia Tech, Blacksburg, VA 24061, USA; 4Present address: Department of Applied Chemistry, Faculty of Engineering, Kyushu University, 744 Motooka, Nishi-ku, Fukuoka 819-0395, Japan

**Keywords:** silica-alumina membrane, dimethyldimethoxysilane (DMDMOS), hydrothermal stability, chemical vapor deposition, gamma-alumina intermediate layers, hydrogen helium separation

## Abstract

Silica-based membranes prepared by chemical vapor deposition of tetraethylorthosilicate (TEOS) on γ-alumina overlayers are known to be effective for hydrogen separation and are attractive for membrane reactor applications for hydrogen-producing reactions. In this study, the synthesis of the membranes was improved by simplifying the deposition of the intermediate γ-alumina layers and by using the precursor, dimethyldimethoxysilane (DMDMOS). In the placement of the γ-alumina layers, earlier work in our laboratory employed four to five dipping-calcining cycles of boehmite sol precursors to produce high H_2_ selectivities, but this took considerable time. In the present study, only two cycles were needed, even for a macro-porous support, through the use of finer boehmite precursor particle sizes. Using the simplified fabrication process, silica-alumina composite membranes with H_2_ permeance > 10^−7^ mol m^−2^ s^−1^ Pa^−1^ and H_2_/N_2_ selectivity >100 were successfully synthesized. In addition, the use of the silica precursor, DMDMOS, further improved the H_2_ permeance without compromising the H_2_/N_2_ selectivity. Pure DMDMOS membranes proved to be unstable against hydrothermal conditions, but the addition of aluminum tri-sec-butoxide (ATSB) improved the stability just like for conventional TEOS membranes.

## 1. Introduction

Silica-based membranes have been known to be effective for selective hydrogen permeation for more than 30 years, and are potentially less expensive than palladium-based membranes. The subject of silica membranes has been covered in recent reviews [1,2,3] and several recent papers [4,5,6,7,8]. Notable past work includes the original papers [9,10], the use of sol-gel techniques [11,12], chemical vapor deposition (CVD) [13,14] and plasma methods [15,16], the employment of diverse CVD precursors [17,18,19,20,21] and compositions [22,23,24], the determination of porosity [25], and the elucidation of the mechanism of permeation [26,27,28]. Important precedents are the use of dimethyldimethoxysilane (DMDMOS) [17] and other methyl-substituted siloxanes [29,30].

The separation mechanism of small species like He, H_2_, and Ne in silica membranes prepared by chemical vapor deposition of alkoxide substrates is based on molecular hopping between solubility sites [31]. The sites in the silica network are approximately 0.3 nm, allowing only small gas molecules like He, H_2_, or Ne to enter them [31,32]. The only way for larger gas molecules to permeate is to pass through pores or defects such as pinholes or cracks. Therefore, the avoidance of defects is important for high H_2_ selectivity, and this will be one of the topics covered in this study. Permeance in membranes with larger pores may occur by a modified gas translational mechanism [33].

Usually, the sol-gel cycle in the synthesis of intermediate layers must be repeated several times to ensure a defect-free membrane [34]. However, too many repetitions not only require longer preparation time with extra energy consumption, but also increase the γ-alumina thickness, thereby posing the risk of cracking. Particularly, a previous study [35] achieved very high H_2_ permeance and selectivity by applying four to five dipping-calcining cycles to form a graded γ-alumina intermediate layer on a macro-porous support, but this was time-consuming and undesirable for reproducibility and cost reasons. It is desirable to reduce the required number of dipping-calcining cycles for a specific membrane support, and this will be one of the topics covered in this study. Scanning electron microscopy (SEM) is used to image samples at intermediate stages of synthesis so as to provide insight on the difference between successful and failed membranes. In addition, the H_2_ permeance is further improved by using a novel silica precursor compound, dimethyldimethoxysilane [17], which makes the silica network slightly looser.

A potential use of silica membranes is in membrane reactors [36,37,38,39,40], so a significant concern is the stability under thermal or hydrothermal conditions [41,42,43]. Particularly, exposure to water vapor at an elevated temperature quickly leads to deterioration of the membrane performance. In general, the H_2_ permeance will decrease due to the densification of the silica network, and the permeance of larger gas molecules will increase due to pinhole generation caused by the degradation of γ-alumina, when it is used as a support. This was confirmed by Reference [17], where hydrothermal stability tests were conducted on both the inside and outside of the membrane tube. The densification of the silica is believed to occur due to the hydrolysis of Si-O-Si linkages and the formation of Si-OH hydroxyl groups, and the degradation of the γ-alumina is believed to occur due to sintering and pore enlargement.

To improve the stability of the silica layer metal elements such as Al [44,45] or Ti [46] can be incorporated into the Si network. They are thought to impart higher tolerance against the attack of the H_2_O molecules to the Si-O-Si bond. For example, in the case of a silica-alumina composite membrane [44], the initial H_2_ permeance was 2–3 × 10^−7^ mol m^−2^ s^−1^ Pa^−1^ with a H_2_/CH_4_ selectivity of 940 at 600 °C, and the H_2_ permeance was maintained above 10^−7^ mol m^−2^ s^−1^ Pa^−1^ even after exposure to 16 mol% steam for 500 h at 600 °C. In the case of a silica-titania composite membrane [46], the initial H_2_ permeance was 3 × 10^−7^ mol m^−2^ s^−1^ Pa^−1^ and was reduced by only 30% after 130 h of exposure in 75 mol% H_2_O at 650 °C, though the H_2_/CH_4_ selectivity was at moderate level (40–60). The above-mentioned membranes are much more stable than pure silica membranes, where a reduction of more than 90% of the H_2_ permeance is typical [47,48]. In this study, silica-alumina composite membranes are employed for hydrothermal stability.

The stability of γ-alumina can also be improved by metal-doping. Silica membranes fabricated by counter diffusion chemical vapor deposition (CVD) of tetramethoxyorthosilicate (TMOS)/O_2_ on pure or Ni-doped γ-alumina showed hydrothermal stability at 500 °C with a steam/N_2_ ratio of 3 [49]. The authors pointed out that for the membrane with pure γ-alumina, the N_2_ permeance fluctuated during the hydrothermal exposure and significantly increased from the initial value, but by doping 5 mol% Ni into the γ-alumina, the problem was minimized. The actual stabilization mechanism, however, was not clarified in the paper. Subsequent work [41] discussed the membrane degradation mechanism in detail, and attributed the fluctuation of the N_2_ permeance to the combination of γ-alumina pore enlargement, pore plugging by grain growth, and subsequent grain rearrangement.

To achieve a high H_2_ permeance without affecting the H_2_ selectivity against the target compound to be separated, it is important to strictly control the pore size of the silica network. Silica membranes fabricated by counter diffusion CVD of TMOS, and one-sided diffusion CVD of phenyltrimethoxysilane (PTMS) and dimethoxydiphenylsilane (DMDPS) have been investigated [50]. The use of DMDPS resulted in a H_2_ permeance of the order of 10^−6^ mol m^−2^ s^−1^ Pa^−1^ and a H_2_/SF_6_ selectivity of 6800 at 300 °C, which is comparable to the performance of palladium membranes. Subsequent work [51,52] using this membrane confirmed that the H_2_/toluene separation ability was stable for more than 1000 h. It should be noted, however, that the H_2_/N_2_ selectivity was lower than ~100 due to an open porous silica network (0.3–0.47 nm [51]) compared to the TEOS- or TMOS-derived networks (~0.3 nm). Several studies have focused on silica membranes with other functional groups like vinyltriethoxysilane [53]. Noteworthy are recent studies that describe high-performance silica membranes [54,55,56].

Silica membranes prepared by counter diffusion CVD have also been studied using a number of silica precursors [57]. Among them, the use of DMDMOS resulted in a superior H_2_ permeance of 9.0 × 10^−7^ mol m^−2^ s^−1^ Pa^−1^ and H_2_/N_2_ selectivity of 920 at 500 °C. However, the paper [17] did not report detailed information about long-term stability or the gas permeation mechanism of the new membrane, so these will be covered in this study.

## 2. Materials and Methods 

### 2.1. Materials

The membranes used in this research consisted of three layers: a commercial α-alumina support (Noritake Co.) with nominal pore size of 60 nm, a γ-alumina intermediate layer deposited by sequential dip-coating/calcining, and a topmost silica layer placed by chemical vapor deposition (CVD). Before silica deposition, two membranes were synthesized in parallel under the same conditions (Figure 1). Then, one membrane was used for CVD and the other membrane was used for scanning electron microscopy (SEM) analysis to have identical substrates for clarifying the relationship between the surface or cross-sectional morphology and the gas separation ability.

### 2.2. Membrane Fabrication

#### 2.2.1. Preparation of the Dipping Sols

The γ-alumina layers were prepared by sol-gel coating of boehmite sols, which were synthesized through the hydrolysis of aluminum isopropoxide (Aldrich, >98%) and subsequent peptization by nitric acid, as reported previously [58]. The boehmite particle sizes were controlled by the time of hydrolysis and the amount of nitric acid. The actual dipping solutions were prepared by mixing the boehmite sols with a polyvinyl alcohol (PVA, Polysciences, M.W. = ~78,000) solution to minimize the risk of cracking by slightly raising the sol viscosity. The obtained concentration of the sol and PVA were 0.15 wt.% and 0.35 wt.%, respectively.

#### 2.2.2. Preparation of the Intermediate Layers

First, a commercial α-alumina tube (i.d. = 4 mm, o.d. = 6 mm, length = 30 mm) with 60 nm pores (Noritake Corporation) was connected to non-porous alumina tubes (i.d. = 4 mm, o.d. = 6 mm, length = 200 mm) at both ends with ceramic joints, which were made by applying a glass paste and firing at 1000 °C. Second, the tube was dipped into the dipping solution for 10 s with its outer surface wrapped with Teflon tape. Third, the tube was dried in a laminar flow enclosure with high efficiency particulate air filters for 4 h, and was calcined at 650 °C for 3 h. Here, a relatively slow heating/cooling rate of 1.5 °C/min was applied in order to minimize the risk of cracking due to different thermal expansion coefficients between the support and the intermediate layer. The dipping-calcining cycle was repeated as necessary. 

#### 2.2.3. Preparation of the Topmost Silica Layers

The topmost silica layers were synthesized by chemical vapor deposition (CVD), which placed a thin (20~30 nm) silica layer on top of the γ-alumina substrate by the thermal decomposition of the silica precursor compound. First, a conventional silica precursor, tetraethyl orthosilicate (TEOS) (Aldrich, 98%), was used to concentrate on the optimization of the intermediate layer. The schematic diagram of the CVD apparatus is shown in Figure 2a. The 6 mm diameter support tube was coaxially fixed inside the stainless reactor using machined Swagelok fittings with Teflon ferrules. The assembly was placed in an electric furnace and was heated to 650 °C at a heating rate of 1.5 °C/min. The bubbler temperature for the TEOS delivery was kept at 25 °C, which gave a TEOS vapor pressure of 250 Pa (obtained from the following Antoine equation).
(1)$$\log_{10}P=A-\frac{B}{T+C},\;A=4.17312,\;B=1561.277,\;C=-67.572,\;P\left[\mathrm{bar}\right],\;T\left[K\right]$$

To improve the hydrothermal stability, a silica-alumina composite membrane was fabricated using a second bubbler to deliver aluminum tri-sec-butoxide (ATSB) as a secondary component in addition to the TEOS (Figure 1a). In this case, the bubbler temperature for the ATSB delivery was set at 96 °C so that the Al/Si ratio would be 0.03. This value was determined by considering the tradeoff between the hydrothermal stability and the H_2_ selectivity obtained in a previous study [44]. The two carrier gases were premixed with a dilution Ar gas before introduction to the inside of the support tube. The lines leading to the inner tube were heated by a ribbon heater to prevent condensation of the precursor compounds. A balance argon gas was also introduced to the outer shell in order to maintain pressure balance between inside and outside of the support tube and minimize the loss of H_2_ permeance due to silica deposition within the pores of the γ-alumina. After the CVD process was finished, the assembly was purged with the balance and dilution Ar gases for about 15 min to sweep out any unreacted precursor compounds or reaction byproducts.

To further improve the H_2_ permeance without compromising the H_2_ selectivity, a novel silica precursor reported in the literature, dimethyldimethoxysilane (DMDMOS) [17], was also examined. The reported H_2_ permeance and H_2_/N_2_ selectivity were 9.0 × 10^−7^ mol m^−2^ s^−1^ Pa^−1^ and 920, respectively, but the authors did not describe any detailed information about the gas permeation mechanism or the long-term stability of the membrane. Therefore, these will be investigated in this study. 

To synthesize DMDMOS membranes, both counter-diffusion CVD (Figure 2b) and one-sided diffusion CVD (Figure 2c) were applied, whereas the previous study [17] dealt with only counter-diffusion CVD. Particularly, the one-sided diffusion CVD made it easier to add the stream of ATSB to improve the hydrothermal stability. In both cases, the CVD temperature was set at 500 °C in order to prevent support cracking due to the large reaction heat between DMDMOS and O_2_ in accordance with the previous report [17].

### 2.3. Evaluation of the As-Synthesized Membranes

#### 2.3.1. SEM Analysis

The surface and cross-sectional microstructures of the membranes were studied using field emission scanning electron microscopy (FESEM, S-900, Tokyo, Japan). Samples were obtained by mechanically slicing the tubular membranes with a diamond saw and subsequently chopping the sliced discs with a cutter. The samples were sputtered with platinum for 15 s before the measurements with SEM. Through the observation, the following points were examined (Table 1).

#### 2.3.2. Single Gas Permeation Test

The CVD process was interrupted at various deposition times, and gas permeation tests were carried out at 650 °C using the same apparatus. The sample gases (H_2_, He, Ne, N_2_, CO_2_, Ar) were introduced into the inside of the membrane tube, the bottom end of which was closed. Then, the gas permeation rates from the outer shell were measured either by a flow meter or a gas chromatograph. Permeances were calculated using the following equation:(2)$$\bar{P}=\frac{F}{A\Delta p}$$
where $\bar{P}$ is the permeance (mol m^−2^ s^−1^ Pa^−1^), F is the molar flow rate (mol s^−1^), A is the permeation area (m^2^), and Δp is the pressure difference across the membrane (Pa). The permeation area, A, was calculated by:(3)$$A=\frac{\pi L\left(r_{1}-r_{2}\right)}{\ln\left(\frac{r_{1}}{r_{2}}\right)}$$
where L is the effective membrane length, $r_{1}$ is the outer diameter, and $r_{2}$ is the inner diameter. The ideal selectivity was defined as the ratio of the single gas permeances:(4)$$\alpha_{ij}=\frac{{\bar{P}}_{i}}{{\bar{P}}_{j}}$$
where $\alpha_{ij}$ is the selectivity of gas i versus j, ${\bar{P}}_{i}$ is the permeance of gas i (mol m^−2^ s^−1^ Pa^−1^), and ${\bar{P}}_{j}$ is the permeance of gas j (mol m^−2^ s^−1^ Pa^−1^). For permeances higher than 10^−9^ mol m^−2^ s^−1^ Pa^−1^, an electronic mass flow meter was used for the measurement, and for lower permeances, a Shimazu 8A gas chromatograph was employed. In the latter case, He gas at a known flow rate was also introduced to the outer shell in order to sweep the permeated molecules to the detector of the gas chromatograph. Then, the gas flow rate was calculated from the He flow rate and sample gas concentration, which was obtained from the average peak area and the calibration curve. The selectivity was calculated as the ratio of the single gas permeances of H_2_ to N_2_, CO_2_, or Ar.

#### 2.3.3. Hydrothermal Stability Test

As was described in the introduction (Section 1), both the topmost selective layer and its support layer must be substantially stable under the operating conditions; otherwise, it is impossible to use the membrane in practical applications. However, there are limited numbers of publications dealing with hydrothermal stability tests on both sides of the membrane. Therefore, in this study, hydrothermal stability tests targeting both the silica and the γ-alumina layers were conducted at 650 °C, up to 200 h.

First, an Ar flow at 6.6 μmol s^−1^ (flow rates in μmol s^−1^ can be converted into ml (NTP) min^−1^ by multiplying by 1.5) was passed through a heated bubbler containing distilled water and was then introduced to the inside of the membrane tube to directly contact the silica layer. The bubbler temperature was set at 56 °C so that the Ar stream would contain 16 mol% water vapor. At the same time, another Ar flow at 11.6 μmol s^−1^ was introduced to the outside of the membrane tube. Both sides of the membrane were kept at atmospheric pressure. The hydrothermal treatment was interrupted at several exposure times, and the permeances of H_2_, He, and N_2_ were measured to monitor the changes in the membrane performance. To make the permeance measurements, the humidified Ar was changed to a pure Ar for about 20 min to dry the membranes. The wet Ar flow was resumed immediately after the permeance measurements. The cycle was repeated several times until the membrane performance became almost unchanged.

Next, the configuration was altered. An Ar flow at 11.6 μmol s^−1^ containing 16 mol% water vapor was introduced to the outer shell to directly contact the γ-alumina layer, while another dry Ar stream at 6.6 μmol s^−1^ was introduced to the inner tube. The gas permeances were measured after several exposure times just like the previous case. Figure 3 summarizes the experimental procedures for the above-described hydrothermal stability tests. As a note, the lines for the delivery of humidified Ar were maintained above 100 °C using ribbon heaters to prevent condensation of water vapor.

## 3. Results and Discussion

### 3.1. Morphology and Structure of the Membranes

Studies with scanning electron microscopy (SEM) revealed a progression of structure in the different membranes (Figure 4). The surface image of the α-alumina support with nominal pore size of 60 nm (Figure 4a) treated once with a 200 nm sol shows that although a certain amount of γ-alumina was deposited on the α-alumina, there were large areas of uncovered α-alumina particles (Figure 4b). The image of the α-alumina support treated by a 200 nm sol twice shows that the α-alumina was totally covered by γ-alumina (Figure 4c). These results mean that at least two dipping-calcining cycles are required to eliminate large defects if large sols are used first. In addition, the support had a coarse surface and relatively large pores (>5 nm) (Figure 4d), so additional treatment with a smaller sol was necessary, thus bringing the total number of dipping-calcining cycles to three.

The γ-alumina surface prepared by dip-coating the α-alumina support with a 40 nm sol (Figure 4e) and 80 and 40 nm sols (Figure 4f) shows the successful formation of smooth and uniform γ-alumina layers. In addition, their cross-sectional images show that the thickness of the γ-alumina layer was about 1 μm for the former (Figure 4g) and about 2 μm for the latter (Figure 4h). These results suggest that macro-porous α-alumina can be completely covered by γ-alumina even if the boehmite particle size is smaller than, or similar to, the support pore size. This has not been reported before.

To further investigate the microstructural differences between these two samples, pictures at higher magnification were taken (Figure 5). Careful observation reveals a slightly higher porosity for the sample with one treatment with 40 nm particles (Figure 5a) than that with successive 80 and 40 nm particles (Figure 5b). Then, the particle stacking density is slightly lower for the former. However, it was difficult to confirm a significant difference even with the nearly maximum resolution of SEM. Earlier studies [54,55,56] have demonstrated the usefulness of nanopermporometry in quantitating the pore size distribution of intermediate layers. Unfortunately, this was not undertaken in the present study. However, the microscopy does indicate that when only one intermediate layer was applied, the surface was only partly covered. 

Although there were no clear differences between the samples before CVD, significant differences were found for the samples after CVD. For the sample with one layer of 40 nm sol, the formation of a silica layer was not complete and there were many visible pinhole defects (Figure 5c), with those at the center of the picture being especially conspicuous. For the sample with successive layers of 80 and 40 nm sols, the surface was formed from continuous rounded structures, which is characteristic of a CVD-derived silica layer (Figure 5d) [59]. In addition, its cross-sectional image revealed that the thickness of the silica layer was about 30–40 nm (Figure 5e), which was similar to that in the previous study. These results clearly show that a smooth and uniform silica layer was successfully formed, although the dipping-calcining cycle was performed only twice on the macro-porous support. As a summary of all the above findings, at least two dipping-calcining cycles are necessary in order to obtain a pinhole-free membrane.

### 3.2. Gas Permeation Properties of the Membranes

Figure 5 shows the changes in gas permeances during CVD for various types of membranes fabricated in this research. At the beginning of the study, a conventional silica precursor, TEOS, was applied to concentrate on the optimization of the intermediate layer. For the membrane with one treatment with a 40 nm sol, the H_2_ permeance at 650 °C was 1.2 × 10^−7^ mol m^−2^ s^−1^ Pa^−1^ and the H_2_/N_2_ selectivity was 26 after 120 min of CVD (Figure 6a). This low selectivity is resonant with the defective surface structure revealed by SEM (Figure 5c). On the other hand, for the membrane successively treated with 80 nm and 40 nm sols, the H_2_ permeance was the same (1.2 × 10^−7^ mol m^−2^ s^−1^ Pa^−1^), but the H_2_/N_2_ selectivity was 330 after 90 min of CVD (Figure 6b). This result clearly demonstrated that a defect-free membrane (Figure 5d) can be synthesized with only two dipping-calcining cycles in spite of the use of a macro-porous 60 nm α-alumina support. Previous studies had used supports with substrate pores of 5 nm [28,35,44]. Therefore, the repetition number of the dipping-calcining cycle was fixed at two for the subsequent studies.

Figure 6c shows the results for one of the successfully prepared membranes by the dual-element CVD of TEOS and ATSB. After 105 min of deposition, the H_2_ permeance and the H_2_/N_2_ selectivity at 650 °C were 2.5 × 10^−7^ mol m^−2^ s^−1^ Pa^−1^ and 980, respectively. The original purpose of the addition of ATSB was to improve the hydrothermal stability (described later), but the H_2_ permeance was also improved compared to the TEOS-derived membranes.

To further improve the H_2_ permeance, the silica precursor was changed from TEOS to DMDMOS. Figure 6d shows the results for the membrane prepared by counter diffusion CVD of DMDMOS and O_2_. In this case, the membrane fabrication was mostly finished within the first 15 min. After that, the H_2_ permeance became almost constant, whereas the N_2_ permeance still decreased slowly, probably because of the modification of some small remaining pores. It is evident that thermal decomposition of DMDMOS was almost negligible, and it actually needed the presence of O_2_ to form a SiO_2_ film. After 60 min of CVD, the H_2_ permeance and the H_2_/N_2_ selectivity at 500 °C were 3.3 × 10^−7^ mol m^−2^ s^−1^ Pa^−1^ and 82, respectively. Considering the fact that the measurement was taken at 500 °C, the H_2_ permeance was already much higher than that for a TEOS-derived membrane, but it was not as high as the literature value (9.0 × 10^−7^ mol m^−2^ s^−1^ Pa^−1^) [17]. Figure 6e shows the results for the membrane prepared by one-sided diffusion CVD of DMDMOS and O_2_. After 45 min of CVD, the H_2_ permeance and the H_2_/N_2_ selectivity at 500 °C were 3.8 × 10^−7^ mol m^−2^ s^−1^ Pa^−1^ and 90, respectively. Finally, Figure 6f shows the results of the membrane prepared by one-sided diffusion CVD of DMDMOS/ATSB/O_2_. After 45 min of CVD, the H_2_ permeance and the H_2_/N_2_ selectivity at 500 °C were 4.0 × 10^−7^ mol m^−2^ s^−1^ Pa^−1^ and 130, respectively.

Overall, the H_2_ permeances of the DMDMOS-derived membranes were not as high as that reported in the literature (9.0 × 10^−7^ mol m^−2^ s^−1^ Pa^−1^) [17], probably due to the lack of optimization of the CVD parameters. However, the permeance measurements for several gas species (Figure 7a) revealed that the DMDMOS-derived membrane may have a looser silica network compared to the TEOS-derived membranes. The first evidence for this inference is that the He/H_2_ permeance ratio for the fresh DMDMOS membrane was as low as 1.2. The value was much lower than those for TEOS-derived membranes (usually >2.0), suggesting that the average pore size of the DMDMOS-derived silica network was significantly larger than the kinetic diameters of H_2_ (0.289 nm) or He (0.26 nm). In fact, the H_2_ permeance was higher than that for a TEOS-derived membrane, but the He permeance was lower. 

The data for the different gases (Figure 7a) can be analyzed by a normalized Knudsen-based permeance method [20]. The method is derived from the gas translation model. First, the gas permeances of the various gases with different kinetic diameters were converted to normalized Knudsen permeances using Equation (5). The obtained results were then plotted as a function of molecular size (Figure 7b). Finally, Equation (6) was used to fit the experimental data, using d_p_ as fitting parameter.
(5)$$f=\frac{\bar{P_{i}}}{{\bar{P}}_{He}\;\sqrt{\frac{M_{He}}{M_{i}}}}$$
(6)$$f=\frac{{\left(1-\frac{d_{i}}{d_{p}}\right)}^{3}}{{\left(1-\frac{d_{He}}{d_{p}}\right)}^{3}}$$

In these equations, where *f* represents the ratio of the permeance of the *i*-th component ($\bar{P_{i}}$) to that predicted from a reference component (He) based on the Knudsen diffusion mechanism, *d_i_* is the kinetic diameter of gas *i* (nm), *d_He_* is the kinetic diameter of helium and *d_p_* is the estimated membrane pore size (nm). P¯He MHeMi is the permeance of the *i*-th component predicted from the He permeance under the Knudsen diffusion mechanism. *M_He_* and *M_i_* are the molecular weight of helium and gas *i*, respectively (g mol^−1^). The analysis gives a pore size of 0.39 nm (Figure 7b). A comparable pure SiO_2_ membrane derived from TEOS gives a pore size of 0.34 nm [53] to 0.37 nm [60], confirming that the structure of the DMDMOS-derived membrane is looser. 

The second evidence for the more open structure of the DMDMOS-derived membrane over a pure TEOS-derived membrane is that the permeance order of CO_2_, Ar, and N_2_ followed the order of their molecular size rather than their molecular weight. This behavior can hardly be observed for a TEOS-derived membrane, where larger gas molecules permeate through defects in the membrane by the Knudsen mechanism, thereby resulting in a permeance order dominated by their molecular weight. The molecular sieving effect for CO_2_, Ar, and N_2_ strongly suggests that there actually were some micropores present which allowed their passage. Furthermore, although the permeances of the above three gas species were on the order of 10^−9^ mol m^−2^ s^−1^ Pa^−1^, the permeance of SF_6_ was on the order of 10^−10^ mol m^−2^ s^−1^ Pa^−1^. This indicated that the membrane had few defects such as pinholes or cracks, and that CO_2_, Ar, and N_2_ permeated through the membrane mainly by the molecular sieving mechanism, and the contribution of the Knudsen permeation was not important. Considering the above findings, there would be a chance to further improve the H_2_ permeance of the DMDMOS membranes by minimizing the membrane thickness through the optimization of the CVD parameters.

### 3.3. Hydrothermal Stability of the Membranes

First, a long-term hydrothermal stability test was conducted for both the inside and outside of the membrane tube using a membrane prepared from TEOS and ATSB. The result is shown in Figure 8. When the inner tube was exposed to 16 mol% water vapor, the H_2_ permeance was rapidly lost for the first 12 h, and then became somewhat stabilized after 96 h of exposure. The tendency was quite normal, and the loss of permeance was much smaller than that for pure silica membranes. The N_2_ permeance remained almost unchanged. Next, the configuration of the apparatus was altered and the outer shell was exposed to 16 mol% water vapor. The original purpose of this treatment was to examine the degree of pinhole enlargement caused by the deterioration of γ-alumina. 

However, unlike the case of membranes prepared by counter-diffusion CVD [41,49], there was no remarkable change in the N_2_ permeance. This was probably due to the structural difference between the membranes prepared by counter-diffusion CVD and one-sided diffusion CVD. In the former case, the silica film is generally deposited within the pores of γ-alumina, thus the continuity of the silica might be susceptive to the structural change of γ-alumina. In the latter case, a thin silica “layer” is deposited on top of the γ-alumina layer, so pinholes can hardly be created even if the average Kelvin pore diameter of γ-alumina increases by the hydrothermal exposure. Rather than a change of the N_2_ permeance, a sharp decrease of the H_2_ permeance was observed again for the first 12 h of exposure. This was probably because of the attack of the H_2_O molecules on both sides of the silica layer, as shown in Figure 9. 

Although the reported kinetic diameter of H_2_O has some variation in the literature (0.265 nm [32], 0.2995 nm and 0.314 nm [61]), the H_2_O permeance is expected to be more than two magnitudes lower than the H_2_ permeance according to the literature [57]. Therefore, it is likely that only the silica network near the surface underwent densification during the first hydrothermal treatment to the inner tube. Then, the silica network near the intersection between the silica and γ-alumina underwent densification during the treatment of the outer shell. Another remarkable finding was that the He/H_2_ permeance ratio tended to increase by exposure to hydrothermal conditions, which has also been pointed out in the literature [41]. This is also indicative of the densification of the silica layer. The kinetic diameters of He and H_2_ are 0.26 and 0.289 nm respectively, whereas the pore size of the TEOS-derived membrane right after synthesis is roughly estimated to be 0.3 nm. As the densification proceeds, the pore size and the H_2_ molecular size will become extremely close to each other, but there still will be a relatively large difference between the pore size and the He molecular size. This is why the reduction in the He permeance was not as sharp as that in the H_2_ permeance.

Next, a hydrothermal stability test was conducted for a pure silica membrane synthesized from DMDMOS, and the result is shown in Figure 10. It was revealed that, although the DMDMOS-derived membrane had better initial performance than a TEOS-derived membrane, it had extremely poor hydrothermal stability and could not maintain its superior performance over a long period of time. The H_2_ permeance before the hydrothermal exposure was 3.2 × 10^−7^ mol m^−2^ s^−1^ Pa^−1^ at 650 °C, and it was reduced by about 75%, and that after 192 h of exposure was 7.8 × 10^−8^ mol m^−2^ s^−1^ Pa^−1^. Another remarkable finding was that the N_2_ permeance also underwent a very large drop from the order of 10^−9^ mol m^−2^ s^−1^ Pa^−1^ to 10^−10^ mol m^−2^ s^−1^ Pa^−1^ by the hydrothermal exposure. This strongly suggests that the micropores through which N_2_ could permeate had shrunk into much smaller micropores that no longer allowed the permeation of N_2_. 

To obtain a deeper understanding of the gas permeation mechanism of this new membrane, the temperature dependence of the gas permeances after the hydrothermal treatment was investigated and the result is shown in Figure 11. The gas permeation behavior after the densification was quite similar to that for a TEOS-derived membrane, probably because the average pore size of the silica layer became about 0.3 nm or smaller by the hydrothermal exposure. It is remarkable that, although the permeance of CO_2_ was higher than that of N_2_ before the exposure, their permeance order became reversed after the exposure, indicating that the permeation mechanism changed from molecular sieve to Knudsen. In any case, a membrane with H_2_ permeance on the order of 10^−8^ mol m^−2^ s^−1^ Pa^−1^ is not attractive for membrane reactor applications as a minimum permeance of 10^−7^ mol m^−2^ s^−1^ Pa^−1^ is needed [40], and therefore, improvement was necessary. 

Then, a hydrothermal stability test was conducted for a silica-alumina composite membrane synthesized by the one-sided diffusion CVD of DMDMOS/ATSB/O_2_, and the result is shown in Figure 12. It can be seen that the hydrothermal stability of the membrane was significantly improved. The initial H_2_ permeance was 4.0 × 10^−7^ mol m^−2^ s^−1^ Pa^−1^, and that after 94 h of exposure became 2.5 × 10^−7^ mol m^−2^ s^−1^ Pa^−1^. The reduction was only 38% in this case. This new membrane was used for the membrane reactor studies on the ethane dehydrogenation reaction in a forthcoming study.

In order to make a comparison, surface SEM pictures after the long-term hydrothermal stability tests were taken for both a defect-free membrane and a deteriorated membrane. Figure 13a shows the surface of a membrane which still had H_2_/N_2_ selectivity above 100. Although the silica surface shows a granular morphology compared to that of a fresh surface (Figure 5d), the γ-alumina was still completely covered by a continuous silica layer. On the other hand, Figure 13b shows the surface of a different membrane which had H_2_/N_2_ selectivity of around only 10 after the hydrothermal exposure. There are many clearly visible pinhole defects, through which larger gas molecules such as N_2_ or CO_2_ could permeate by Knudsen diffusion. 

Figure 14 shows a summary of the performance of a total number of 16 membranes fabricated in this study, the intermediate layers of which were synthesized by the sequential treatment with an 80 and a 40 nm sol. Even though the synthesis steps were simplified compared to a previous study [35], Si-Al composite membranes with H_2_ permeance > 1.5 × 10^−7^ mol m^−2^ s^−1^ Pa^−1^ and H_2_/N_2_ selectivity > 100 have been successfully fabricated many times, demonstrating the level of reproducibility of the membrane fabrication process used in this study. There are many steps that enter into the synthesis, including joining of the support to the dense alumina tubes, the preparation of the sols, the dip-coating procedure, and the CVD process, and it is difficult to control all the steps. Nevertheless, the results are significant as the values of H_2_ permeance > 1.5 × 10^−7^ mol m^−2^ s^−1^ Pa^−1^ and H_2_/N_2_ selectivity > 100 noted above were shown to be requirements for a commercial process [36]. This involved the study of the model A → B + C reaction, the dihydrogen-ation of ethane (C_2_H_6_ → C_2_H_4_ + H_2_) over a 5 wt% Cr/ZSM-5 catalyst in a conventional packed-bed reactor (PBR) and in a membrane reactor (MR) fitted with hydrogen-selective silica membranes. Specifically, it was demonstrated that for the all-important permeance range of 10^−7^ and 10^−6^ mol m^−2^ s^−1^ Pa^−1^, there was an enhancement in yield as H_2_ selectivity increased from 10 to 20 and 30, but beyond 100, there was little improvement even with a selectivity of 1000 [36]. Thus, a selectivity of 100 is sufficient for most applications involving membrane reactors.

It should be noted, however, that the membrane performance (especially the H_2_ permeance) tended to be reduced after the long-term hydrothermal treatment, and it was not easy to maintain the above-mentioned performance over a long period of time. It was also remarkable that, when the initial H_2_/N_2_ selectivity was lower than 100, the membrane performance tended to deteriorate by the hydrothermal treatment, probably because of fatal pinhole enlargement. This provides a direction for improvement, that is the number of defects should be reduced at the outset.

## 4. Conclusions

Silica-alumina composite membranes with H_2_ permeance > 1.5 × 10^−7^ mol m^−2^ s^−1^ Pa^−1^ and H_2_/N_2_ selectivity > 100 have been successfully fabricated repeatedly, even though the membrane fabrication steps were simplified compared to previous studies. In addition, the use of a novel silica precursor, dimethyl dimethoxy silane (DMDMOS), further improved the H_2_ permeance without compromising the H_2_ selectivity. Even though the pure silica membranes synthesized from DMDMOS had poor hydrothermal stability just like the TEOS-derived membranes, the addition of aluminum tri-sec-butoxide (ATSB) rendered the membrane much more stable.

## Figures and Tables

**Figure 1 membranes-10-00050-f001:**
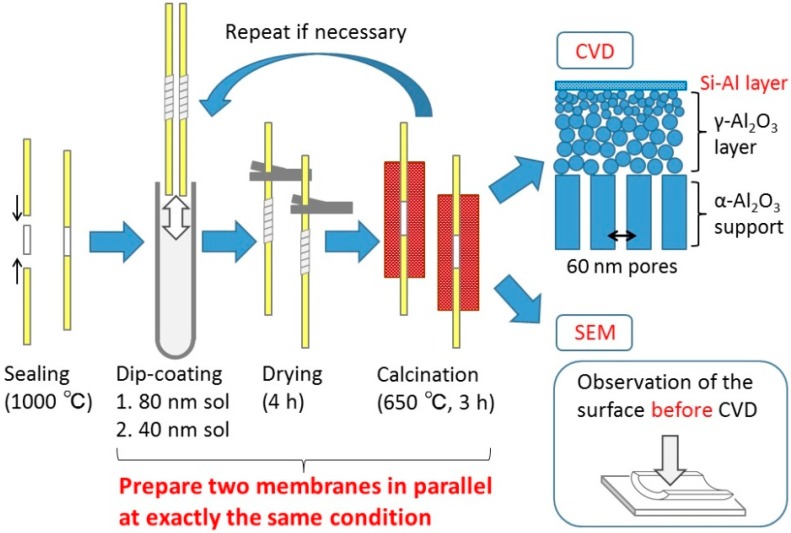
Membrane fabrication procedures. CVD-Chemical vapor deposition, SEM-Scanning electron microscopy.

**Figure 2 membranes-10-00050-f002:**
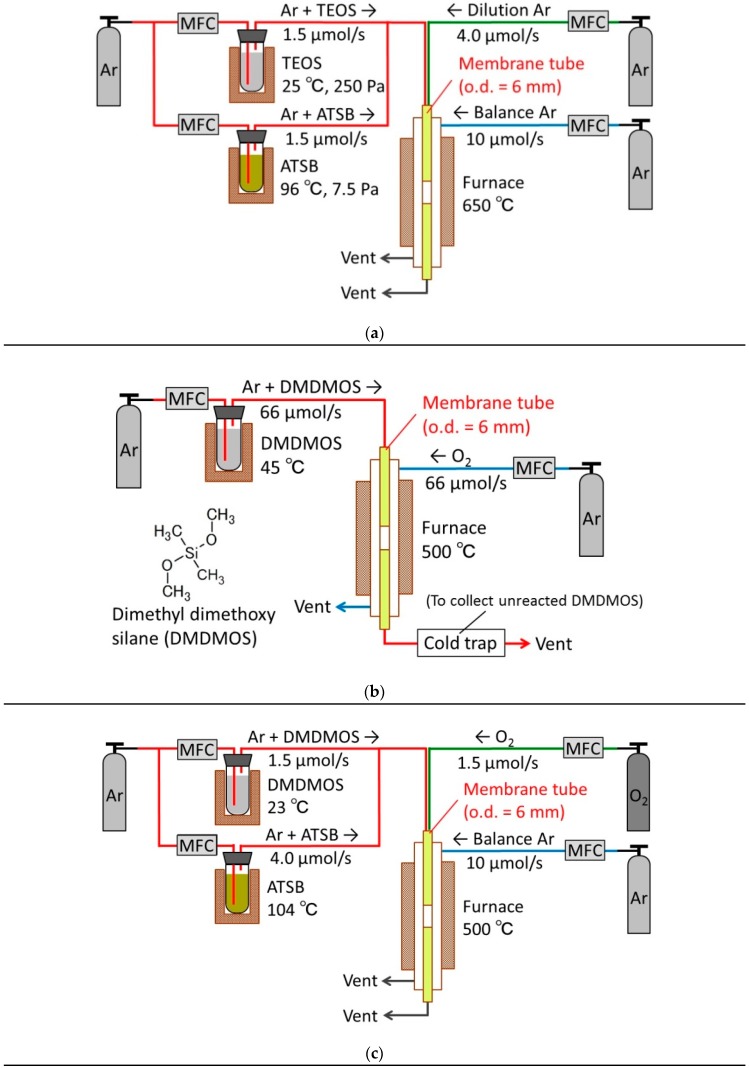
Experimental setup for several types of chemical vapor deposition (CVD). (**a**) One-sided diffusion CVD of tetraethylorthosilicate (TEOS)/aluminum tri-sec-butoxide (ATSB), (**b**) counter-diffusion CVD of dimethyldimethoxysilane (DMDMOS)/O_2_, (**c**) one-side diffusion CVD of DMDMOS/ATSB/O_2_.

**Figure 3 membranes-10-00050-f003:**
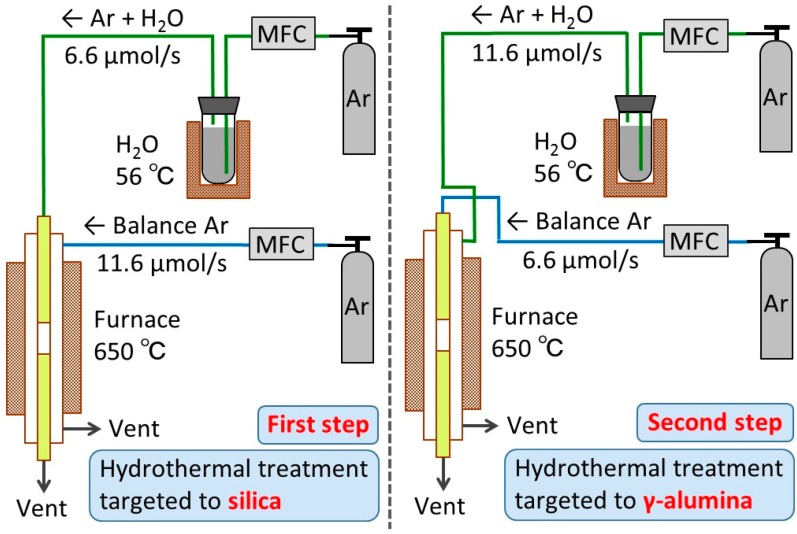
Hydrothermal stability test for both the inside and outside of the membrane tube. MFC-Mass flow controller.

**Figure 4 membranes-10-00050-f004:**
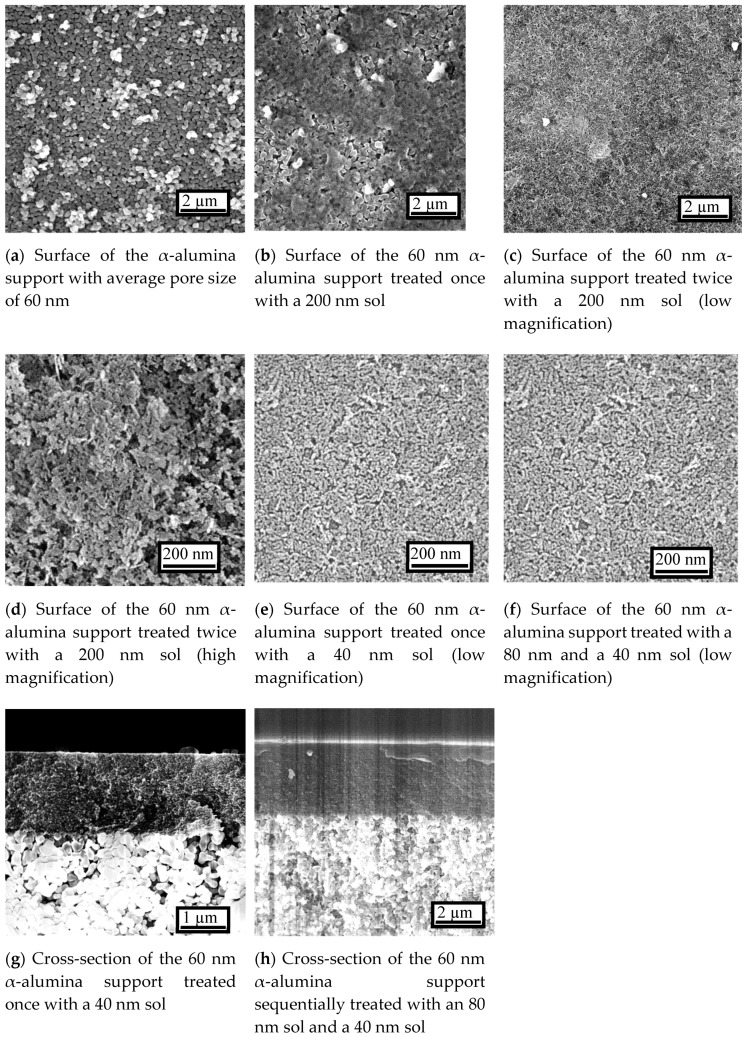
Scanning electron microscopy (SEM) images of the membranes at low magnification.

**Figure 5 membranes-10-00050-f005:**
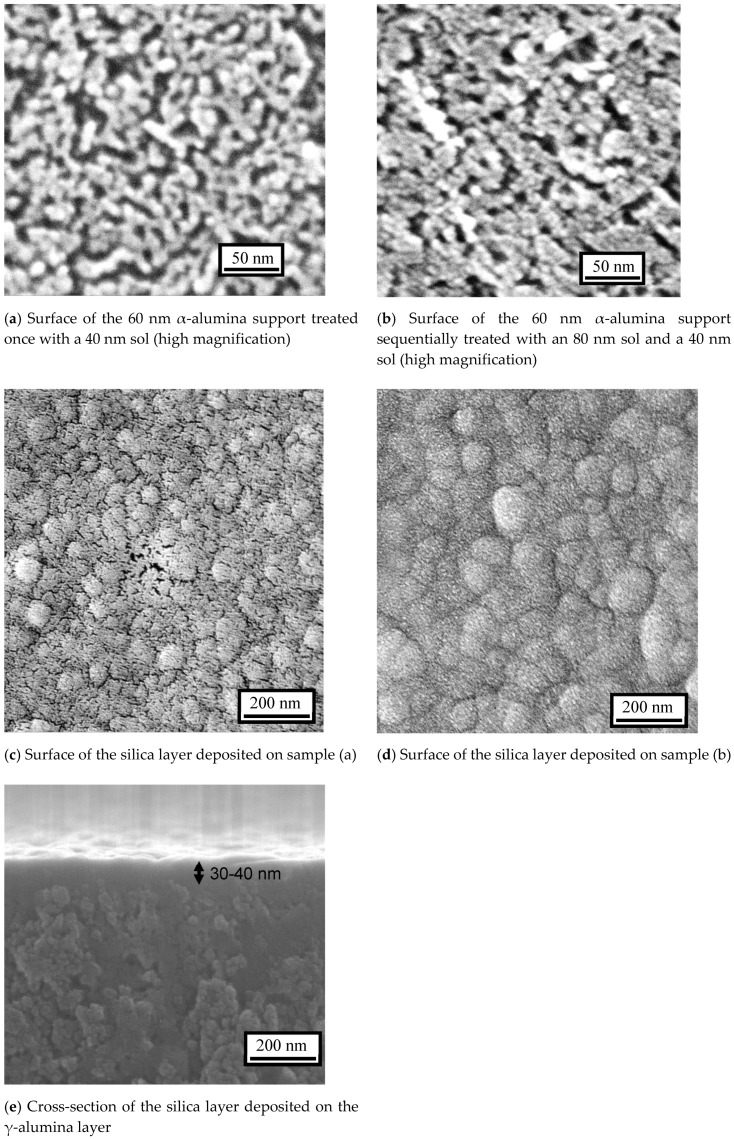
SEM images of the membranes at high magnification.

**Figure 6 membranes-10-00050-f006:**
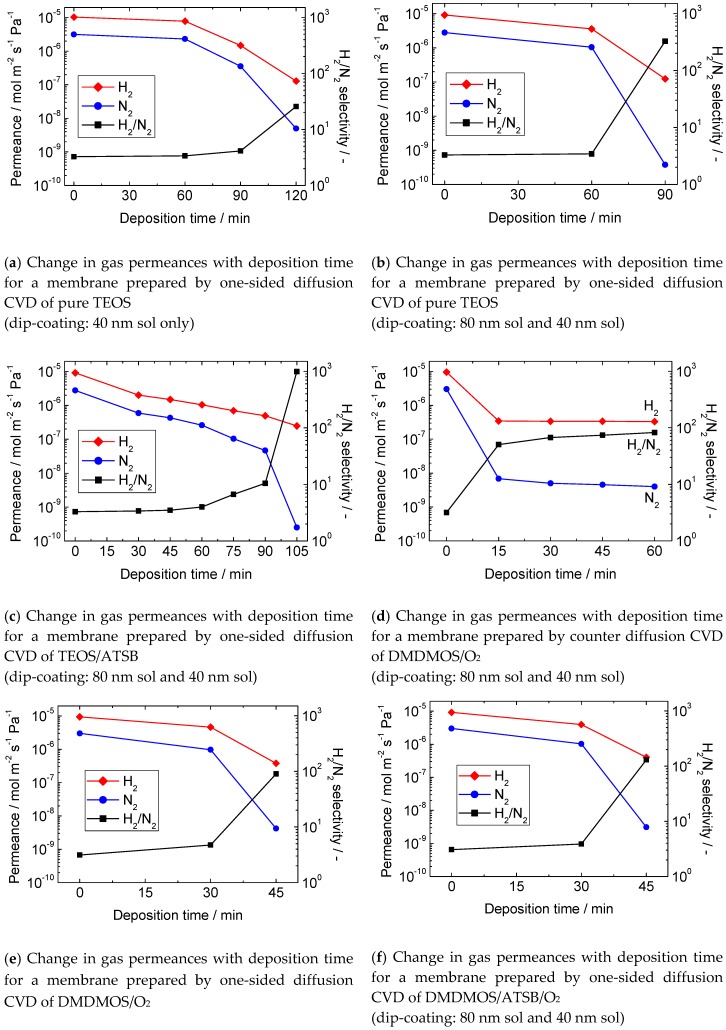
Change in gas permeances with deposition time for various types of membranes.

**Figure 7 membranes-10-00050-f007:**
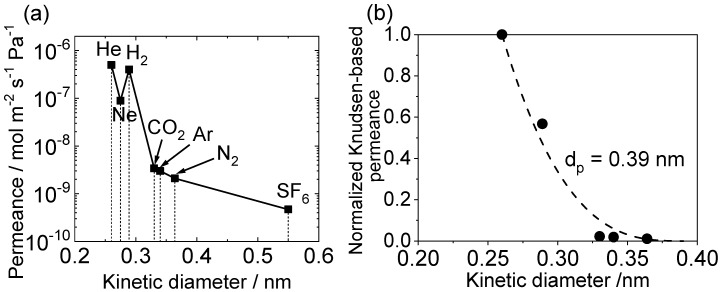
(**a**) Gas permeances for a membrane synthesized by counter-diffusion CVD of DMDMOS and O_2_ measured at 650 °C (before hydrothermal treatment), (**b**) determination of pore size.

**Figure 8 membranes-10-00050-f008:**
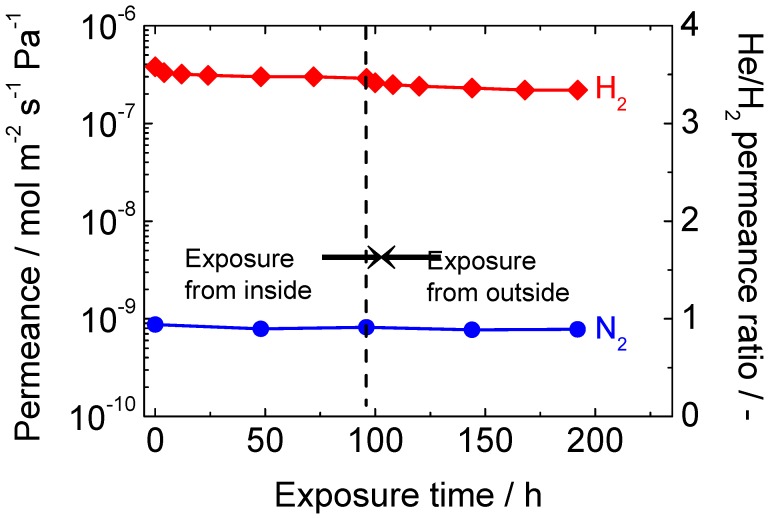
Results of the long-term hydrothermal stability tests to both the inside and outside of the membrane tube.

**Figure 9 membranes-10-00050-f009:**
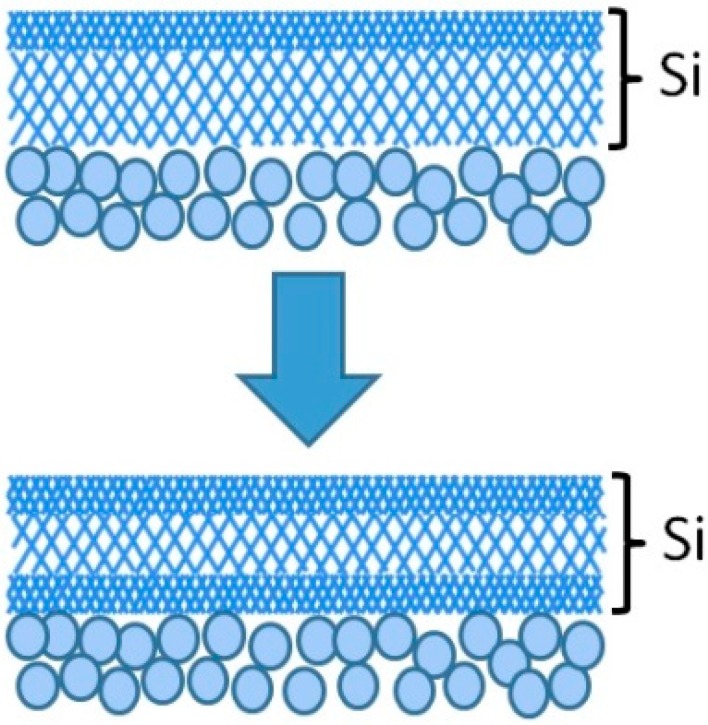
Proposed mechanism for the sharp decrease of the H_2_ permeance, which was observed twice.

**Figure 10 membranes-10-00050-f010:**
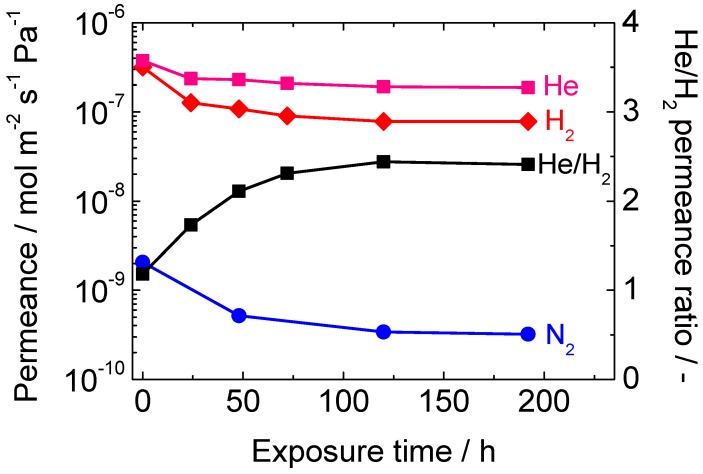
Results of the hydrothermal stability test for a pure silica membrane synthesized from DMDMOS.

**Figure 11 membranes-10-00050-f011:**
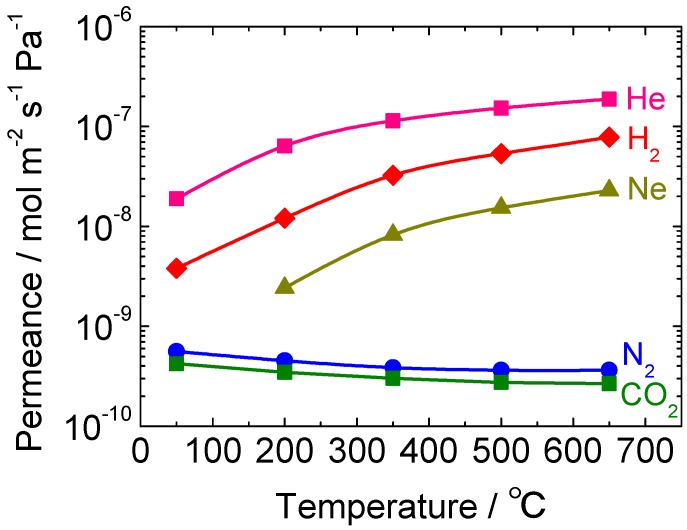
Temperature dependence of the gas permeances after the hydrothermal treatment for a pure silica membrane synthesized from DMDMOS.

**Figure 12 membranes-10-00050-f012:**
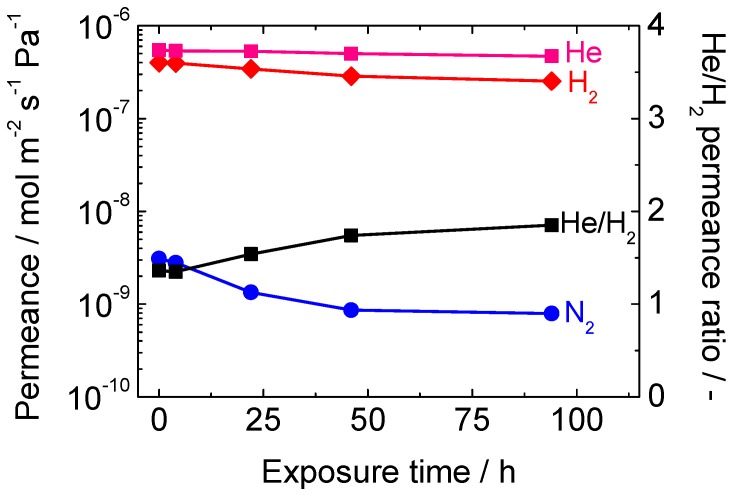
Result of the hydrothermal stability test for a silica-alumina composite membrane synthesized by one-side diffusion CVD of DMDMOS/ATSB/O_2_.

**Figure 13 membranes-10-00050-f013:**
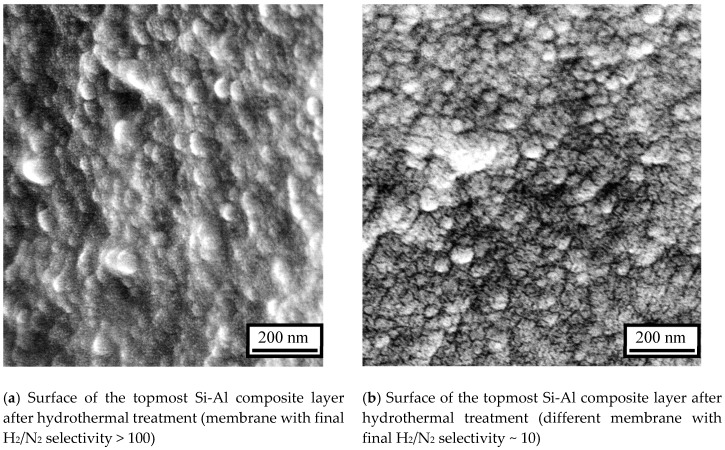
SEM images of the membrane surface after the long-term hydrothermal treatment.

**Figure 14 membranes-10-00050-f014:**
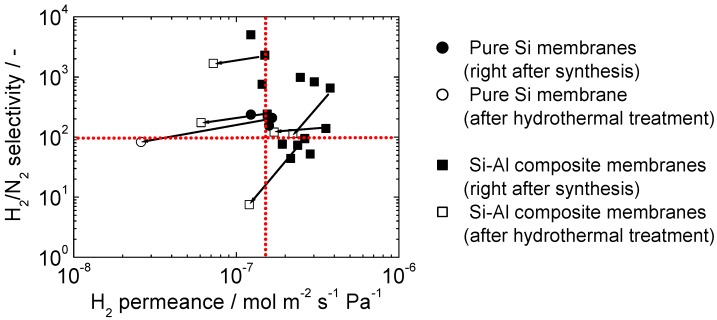
Summary of the performance of a total number of 16 membranes synthesized from TEOS or TEOS + ATSB.

**Table 1 membranes-10-00050-t001:** Characterization of Membranes by Scanning Electron Microscopy.

	Samples before CVD	Samples after CVD
Surface	∙ Particle size of γ-alumina ∙ Porosity ∙ Smoothness ∙ Presence of pinholes	∙ Smoothness ∙ Presence of pinholes
Cross-section	∙ Thickness of the γ-alumina layer ∙ Presence of γ-alumina particle ∙ infiltration into α-alumina ∙ Presence of cracks	∙ Thickness of the silica layer ∙ Presence of silica infiltration ∙ into γ-alumina ∙ Presence of cracks

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
