# Peer review of "Synthesis of Silica Membranes by Chemical Vapor Deposition Using a Dimethyldimethoxysilane Precursor"

_membranes, 2020, doi:10.3390/membranes10030050_

Round 1
Reviewer 1 Report
The manuscript is a study on synthesis of silica-based membranes was improved by simplifying the deposition of the intermediate γ-alumina layers and by using the precursor, dimethyldimethoxysilane (DMDMOS). The paper gives long-term stability or gas permeation mechanism of the new silica-based membranes. but it needs a deep revision of the English language, in addition to minor and major revisions.
- English need a deep revision. There are errors in syntax (especially), grammar, plurals/singular, conjugation of verbs, punctuation.
- Lines 107~108 “[17Error! Bookmark not] “, line 454 “ literature [Error! Bookmark not defined.]”
- The titles of Fig.3 (a), (b), (c), Fig. 5 (a), (b), (c)v are covered on the figures.
- 5 has two (c), which figure is Fig. 4?
- The SEM images is not clear, please give the clear pictures.
- Lines 497 “ exhibits ” should be modified “Fig. 13 exhibits”
Author Response
Please see the attachment. Unfortunately, I could not attach the annotated manuscript for you to see.

Reviewer 2 Report
This paper discusses an improved method for synthesis of γ-alumina intermediate layers and stability and gas permeation mechanism of silica membranes prepared by CVD using a dimethyldimethoxysilane(DMDMOS).
【Comment 1】
The authors tried to clarify the synthesis and gas permeation mechanism of the silica membranes and γ-alumina intermediate layers using SEM observations and single gas permeation tests; however, the mechanism is still unclear, mainly because there seems no information on the pore size distributions of effective transport pathways across the prepared intermediate layers and membranes, especially in the range of 1-50 nm.
Since the thickness of prepared γ-alumina intermediate layers seem 1-2 μm in Fig. 3, only surface SEM images seem to be not enough to evaluate the pore size of effective transport pathway across the membranes. On this point, only single gas permeation tests seem to be not enough because they only characterize the permeation pathway of less than 0.6 nm at best.
(Suggestion)
Addition of nanopermporometry characterization (pore size distribution data for gas permeation) seems to be very helpful for readers to understand the quality and the gas permeation mechanism of the prepared γ-alumina intermediate layers and silica membranes.
【Comment 2】
The authors claimed that γ-alumina intermediate layers were prepared using a simplified fabrication process; however, comparison of quality (such as gas permeability, pore size distribution, and stability) and reproducibility data should be presented at least.
On this point, the authors claimed that Si-Al composite membranes with H2 permeance of > 1.5 x 10-7 [mol/(m2 s Pa)] and H2/N2 selectivity 100 have been successfully fabricated repeatedly; however, In Fig 13, performance of Si-Al composite membranes (right after synthesis) varied largely (H2 permeance of 1-4 x 10-7 [mol/(m2 s Pa)] and H2/N2 selectivity of 50-5,000)
Readers seem to be unclear that the difference of membrane performance is caused by difference in quality of prepared γ-alumina intermediate layers or not.
Explanations on the difference in the performance of membranes fabricated by the same synthesis condition should be presented.
Author Response

(The authors gave the same response as above.)

Round 2
Reviewer 1 Report
The paper has been improved and it may be published in the present form.
Reviewer 2 Report
Thank you for your modification.
The revised manuscript added an micropore analysis on silica membranes; however, gas transport mechanism is still unclear because there seems no information on the gas permeation in the range of 1-50 nm. We cannot discuss the gas leak of N2 and/or SF6 was caused by pinhole (1-50 nm) of γ-alumina intermediate layers and/or CVD process of micropore tuning.
In particular, I have some difficulty in accepting the idea of the sentences of line 20-25 (In the placement of the γ-alumina layers, earlier methods employed four to five dipping-calcining cycles of boehmite sol precursors which took considerable time. In the present study only two cycles were needed even for a macroporous support, through the use of finer boehmite precursor particle sizes. Using the simplified fabrication process, silica-alumina composite membranes with H2 permeance > 10-7 mol m-2 s-1 Pa-1 and H2/N2 selectivity >100 were successfully synthesized.)
If the authors want to claim the improvement of fabrication process of γ-alumina intermediate layers, I would like to suggest additional tests of nanopermporometry experiments for intermediate layers, which other research groups generally employed for characterizing pore size distribution using Kelvin equation.
<Examples of recent literatures where nanopermporometry was used for developing silica membranes>
(Example 1)
'Development of CVD Silica Membranes Having High Hydrogen Permeance and Steam Durability and a Membrane Reactor for a Water Gas Shift Reaction'
Ryoichi Nishida , Toshiki Tago, Takashi Saitoh, Masahiro Seshimo and Shin-ichi Nakao
Membranes 2019, 9(11), 140; https://doi.org/10.3390/membranes9110140
(Example 2)
'Development of Mass Production Technology of Highly Permeable Nano-Porous Supports for Silica-Based Separation Membranes'
Ken-ichi Sawamura * , Shigeru Okamoto and Yoshihiro Todokoro
Membranes 2019, 9(8), 103; https://doi.org/10.3390/membranes9080103
(Example 3)
'Silica-Based RO Membranes for Separation of Acidic Solution'
Katsunori Ishii, Ayumi Ikeda, Toshichika Takeuchi, Junko Yoshiura and Mikihiro Nomura
Membranes 2019, 9(8), 94; https://doi.org/10.3390/membranes9080094
If the authors cannot conduct additional tests in this research and do not stick to claim the improvement of fabrication process of γ-alumina intermediate layers, I would like to suggest some comparison and /or modification of the sentences of line 20-24 because no comparison was shown on the number of dipping-calcining cycles in this paper. If increasing the number of dipping-calcining cycles from 2 to 3 enhance H2/N2 selectivity from 100 to over 1000 and/or reproducibility, the number of dipping-calcining cycles may need to be more than 2.
On this point, is the suggested membrane performance (H2 permeance > 10-7 mol m-2 s-1 Pa-1 and H2/N2 selectivity >100) enough for commercial process ? In other word, is the H2/N2 selectivity over 1000 not necessary for commercial process ? If the required H2/N2 selectivity is only 100, the claim (Line 20-25) can be more acceptable.
Some additional explanation may be helpful for readers to be agree with the idea (Line 20-25) of the authors.
Author Response
Please see attached response.
